# ProRefine: Inference-Time Prompt Refinement with Textual Feedback

## Abstract

Agentic workflows, where multiple LLM based agents collaborate to accomplish complex tasks like reasoning or planning, play a substantial role in many cutting-edge commercial applications, and continue to fascinate researchers across fields for their potential to accomplish expensive, complex tasks that, until recently, only humans have been trusted to do. These workflows critically depend on the prompts used to provide the roles models play in such workflows. Poorly designed prompts that fail even slightly to guide individual agents can lead to sub-optimal performance that may snowball within a system of agents, limiting their reliability and scalability. To address this important problem of inference-time prompt optimization, we introduce ProRefine, an innovative inference-time optimization method that uses an agentic loop of LLMs to generate and apply textual feedback. ProRefine dynamically refines prompts for multi-step reasoning tasks without additional training or ground truth labels. Evaluated on five benchmark mathematical reasoning datasets, ProRefine significantly surpasses zero-shot Chain-of-Thought baselines by 3 to 37 percentage points. This approach not only boosts accuracy but also allows smaller models to approach the performance of their larger counterparts. This highlights its potential for building more cost-effective and powerful hybrid AI systems, thereby democratizing access to high-performing AI.

## 1 Introduction

The rapid evolution of Large Language Models (LLMs) has transformed applications from search engines to automated reasoning systems (Feng et al., 2024). Yet, their efficacy criticality depends on alignment with human values and preferences, a process often bottlenecked by the scarcity and subjectivity of human feedback (Gray & Suri, 2019; Kahneman et al., 2021; Prabhakar et al., 2024; Weerasooriya et al., 2023). LLMs, trained on vast datasets, utilize alignment techniques to generate more human-like and accurate responses (Kirk et al., 2024; Rao et al., 2023; Sorensen et al., 2024).

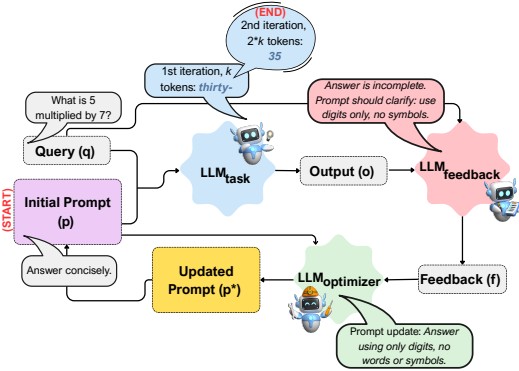

Figure 1: Overview of ProRefine system, illustrating the iterative process of prompt optimization using feedback from LLMs. In each iteration, $LLM_{task}$ extends its output by an additional k tokens, enabling step-by-step feedback to progressively refine the prompt with $LLM_{optimizer}$. Each refinement iteration updates the prompt for future tokens; previous tokens remain unchanged.

Recent advances have shifted toward leveraging LLMs themselves as judges and evaluators Bavaresco et al. (2024); Chiang & Lee (2023); Li et al. (2024); Liu et al. (2023); Wang et al. (2023); Zheng et al. (2023). Building on this, *compound* AI systems that use LLMs to generate feedback / critiques Pryzant et al. (2023); Saunders et al. (2022). Akyurek et al. (2023) and Wadhwa et al. (2024) explore agentic frameworks to detect, critique, and refine tasks that require factual accuracy. Another approach is to iteratively refine the output using self-generated feedback Madaan et al. (2023). TextGrad Yuksekgonul et al. (2024) introduced automatic "differentiation" via text, using textual feedback to optimize the performance of individual components within a compound AI system.

Our work focuses on optimizing the *prompt*, a key element in chain-of-thought (CoT) (Wei et al., 2022) based LLM reasoning. CoT mimics human problem-solving by breaking down complex tasks into smaller, manageable steps. The system we introduce here, **ProRefine** (Inference-time **Pro**mpt **Refine**ment with Textual Feedback), builds upon CoT by adaptively improving prompts using feedback ($LLM_{feedback}$) and an optimizer ($LLM_{optimizer}$) to refine prompts for the task-performing LLM ($LLM_{task}$). This workflow (Figure 1), motivated by the teacher-student framework Torrey & Taylor (2013) where a teacher agent guides a student agent to perform a task by providing feedback at intermediate steps, but implemented via LLM interactions without pre-training, represents a novel approach to adaptive agentic reasoning. While these agents may be instantiated as separate LLMs (potentially with different architectures or parameter sizes), they operate collaboratively in a closed-loop to iteratively refine prompts and outputs. We explore policy optimization for aligning compound AI systems, drawing inspiration from TextGrad and policy gradient algorithms like PPO. This work investigates the following research questions.

**RQ1** *How effectively can textual feedback enhance LLM performance of LLMs during inference?*

**RQ2** *To what extent does model size impact the ability of LLMs to utilize textual feedback?*

**RQ3** *What is the impact of incorporating a verifier on accuracy at inference time?*

Although prior work has explored prompt optimization Deng et al. (2022); Dong et al. (2024); Shin et al. (2020); Yang et al. (2024), they all often focus on either *offline fine-tuning*, which requires extensive training data, or universal application of *largest, most capable models to every task*. This presents a practical dilemma in many real-world scenarios. Continuously fine-tuning is not always feasible, and relying exclusively on state-of-the-art models is often computationally prohibitive. A different approach is needed for scenarios that require *dynamic, on-the-fly repair* for specific and difficult queries where a standard prompt fails. This is particularly true in *resource-aware deployments*, where a smaller model may suffice for most tasks but requires enhancement for a small subset of critical queries. ***The goal, therefore, shifts from finding a single, universally optimal prompt to performing targeted, inference-time intervention.***

This hybrid-model paradigm makes a method like ProRefine a practical solution. It is designed for resource-constrained environments where deploying the largest models for every query isn't feasible, but temporary access to a capable feedback LLM (perhaps via a separate API call) is possible for critical tasks. In such cases, the refinement process is triggered as an on-demand "expert intervention." ProRefine is task-agnostic and requires no additional training or ground-truth labels. It is an inference-time optimization method that relies on the availability of test-time compute and the ability of LLMs to provide and act upon feedback for optimization.

The ability to break complex tasks into smaller steps and dynamically improve prompts offers a crucial advantage in multi-step agentic workflows where errors can compound. As illustrated in Figure 4, this method is also suitable for black-box LLMs where only API access is available. ProRefine could prove to be crucial in situations demanding greater interpretability, where feedback steps (outputs of $LLM_{feedback}$) offer insights into the reasoning correction process and applications requiring dynamic adaptation without retraining/fine-tuning cycles. To demonstrate its effectiveness, we evaluate ProRefine across five benchmark mathematical reasoning datasets, showing it offers a robust alternative to solely scaling up the base model for all queries.

**Key Contributions:**

- We propose a novel method - ProRefine - for prompt optimization *inference-time* using textual feedback.

- We evaluated ProRefine on five datasets: object counting, word sorting, grade-school math problem solving, math word problems, and algebraic word problems, and compared our method against CoT and TextGrad.
- We evaluate the importance of using a verifier at inference time.

Essentially, ProRefine is an elegant, training-free, inference-time solution that fixes the core prompt to improve reasoning, making it suitable for practical hybrid-model deployment.

## 2 RELATED WORK

ProRefine draws inspiration from and contributes to several interconnected research areas. The performance of LLMs is heavily dependent on the quality of the prompts they receive. Early efforts in this domain centered on manual prompt crafting Wei et al. (2022), a meticulous process of designing effective prompts to elicit desired responses. Recognizing the limitations and scalability challenges of manual methods, research has increasingly focused on automatic prompt optimization with a growing emphasis on agentic workflows that enable dynamic and adaptive reasoning.

### 2.0.1 PROMPT GENERATION

Some pioneering automatic methods, such as AutoPrompt Shin et al. (2020) and RLPrompt Deng et al. (2022), employ gradient-based search and reinforcement learning techniques, respectively. AutoPrompt Shin et al. (2020) uses gradient-based search to generate prompts for masked language models. It reformulates tasks as fill-in-the-blank problems, achieving performance comparable to supervised models in tasks like sentiment analysis. However, it requires training data and gradient access, limiting its applicability to black-box models. Other approaches leverage LLMs themselves for prompt generation Mehta et al. (2024); Pryzant et al. (2023); Yang et al. (2024; 2022); Zhou et al. (2022). Recent works like Promptomatix Murthy et al. (2025) and EvoAgentX Wang et al. (2025) extend this direction by enabling automatic prompt refinement across multiple tasks, workflows, and tools. ProRefine distinguishes itself by operating simply at inference-time, requiring no training data, gradient access, or model retraining, while enabling prompt refinement in dynamically evolving settings.

### 2.0.2 SELF-REFINEMENT

There is a substantial and growing body of work exploring the capacity of LLMs to act as judges or evaluators Bavaresco et al. (2024); Chiang & Lee (2023); Li et al. (2024); Liu et al. (2023); Verga et al. (2024); Wang et al. (2023); Zheng et al. (2023); Zhuge et al. (2024). This capability has been leveraged to assess response quality or provide self-feedback. ProRefine adopts this principle, using LLM-generated textual feedback to improve its own prompting process. Unlike prior uses of LLM evaluation solely for ranking or filtering, ProRefine uses that feedback in a closed-loop for optimization during inference.

The idea of LLM iterative refinement is highly relevant. Self-Refine Madaan et al. (2023) is a prominent example, where an LLM generates both output and feedback, using the latter for refinement. ARIES Zeng et al. (2025) further enhances refinement via Elo-style agent debate. Other works explore self-critiquing Saunders et al. (2022) and reinforcement learning for critique generation (RL4F) Akyurek et al. (2023), along with various feedback and refinement mechanisms Dong et al. (2024); Khattab et al. (2024); Qu et al. (2024); Ranaldi & Freitas (2024); Schick et al. (2023); Wadhwa et al. (2024), and Monte Carlo-based refinement in math reasoning (MC-NEST) Rabby et al. (2025). While ProRefine shares the self-refinement spirit, it focuses on prompt refinement, suitable for agentic workflows and black-box LLMs, while avoiding reinforcement learning and direct output modification.

### 2.0.3 INFERENCE-TIME SCALING

ProRefine belongs to the broader category of inference-time methods Muennighoff et al. (2025); Snell et al. (2024a), that improve LLMs without without weight modification Du et al. (2024). Inference-time methods aim to improve the performance of models by utilizing test-time compute resources. TextGrad Yuksekgonul et al. (2024) performs gradient-free inference-time optimization using textual

feedback. ProRefine applies a similar idea to intermediate prompt refinement for dynamic reasoning chains. TextGrad relies on supervised fine-tuning, whereas ProRefine operates without training data, offering ease of integration. Other inference-time strategies include RL-of-Thoughts Hao et al. (2025) and Reward-Is-Enough Song et al. (2025), which apply RL-based signal propagation during inference. AvR (Alignment via Refinement) Zhang et al. (2025a) proposes recursive CoT refinement using long-form reasoning. ProRefine, by contrast, performs step-level feedback on prompts rather than final outputs, and requires no external tools or supervision.

### 2.0.4 AGENTIC WORKFLOWS

ProRefine also fits into a broader trend toward agentic workflows. AFlow Zhang et al. (2025b) automates agentic workflows through prompt-based search over prior structures, while EvoAgentX Wang et al. (2025) evolves agent behaviors and topologies. Meanwhile, Mass Zhou et al. (2025) and DebFlow Su et al. (2025) optimize multi-agent configurations via interleaved search and debate. ProRefine focuses instead on optimizing individual agent prompts within fixed workflows, complementing these methods. Unlike tool-integrated or debate-based systems, ProRefine remains model-agnostic and easy to integrate into any prompt-based agent loop.

## 3 PROREFINE

**ProRefine** is an inference-time prompt optimization algorithm that optimizes prompts by using textual feedback. Each refinement iteration updates the prompt for future tokens; previous tokens remain unchanged. ProRefine involves interactions between three LLMs:

$LLM_{task}$: Executes the task based on the current prompt, generating the initial and subsequent outputs.

$LLM_{feedback}$: A model that critiques the $LLM_{task}$'s output, providing detailed feedback on improvements. This model should be capable of providing insightful and accurate critiques Bai et al. (2022); Saunders et al. (2022).

$LLM_{optimizer}$: Interprets the feedback and refines the prompt, aiming for coherent and task-focused improvements. This LLM is crucial for ensuring the prompt evolves effectively.

**ProRefine**   (Algorithm 1) works as follows:

---
**Algorithm 1:** ProRefine

---
**Input:** Query: $q$, Initial prompt: $p$, tokens_per_step: $k$, max_steps: $n$, LLMs: $LLM_{task}$,
       $LLM_{feedback}$, $LLM_{optimizer}$
**Output:** Optimized prompt: $p^*$
$p^* = p$
**for** $i = 1$ **to** $n$ **do**
  $o_i = LLM_{task}(p^*, q)$ // Generate $i * k$ tokens
  $f_i = LLM_{feedback}(q, o_i)$ // Get textual feedback
  $p* = LLM_{optimizer}(p^*, f_i)$ // Optimize the prompt
  **if** $EOS\_token$ in $o_i$ **then**
    └ **break**
**return** $p^*$ // Return final optimized prompt

---

**Initialization:**   Start with an initial prompt $p$ for the task, a query $q$, and parameters defining the generation and optimization process ($k$ tokens per step, $n$ maximum steps).

**Generation and Feedback Loop:**

- **Generation:** Use $LLM_{task}$ to generate an output based on the current prompt $p^*$ and query $q$. This step is limited to $i * k$ tokens to control the granularity of the feedback. In each iteration, $LLM_{task}$ produces $k$ more tokens, attempting to refine prior output while progressively continuing its response to the query.

- **Feedback:** $LLM_{feedback}$ evaluates the generated output $o_i$ against the query $q$ to provide textual feedback $f_i$. This feedback encapsulates how the output could be improved, focusing on aspects such as accuracy, relevance, or coherence.

- **Optimization:** $LLM_{optimizer}$ uses the feedback $f_i$ to refine the prompt $p^*$. This step involves modifying the prompt to better align with the task requirements or to correct identified deficiencies in previous generations.

**Termination:** The process iterates until either the maximum number of steps $n$ is reached or an end-of-sequence (EOS) token is detected in the output, indicating the completion of the task.

## 3.1 UNIFYING VERIFIER AND FEEDBACK

At inference time, verifiers play a crucial role in judging model outputs Cobbe et al. (2021); Lightman et al. (2024); Snell et al. (2024b). Often trained on extensive corpora, these verifiers can be outcome-based, assessing the final result's correctness, or process-based, evaluating the validity of intermediate steps Lightman et al. (2024); Snell et al. (2024b); Uesato et al. (2022). For simplicity in this study, we do not train a bespoke verifier; rather, we employ the *Llama3.1-70B-instruct* model to function as both the feedback mechanism ($LLM_{feedback}$) and the verifier. We manage these roles through separate API calls, each with a role-defining prompt. A smaller model, specifically fine-tuned for these tasks, could also be used. The verifier's function is to evaluate the initial output generated by $LLM_{task}$ for each query and to halt generation early on. If verifier assess this output is incorrect, the refinement process is triggered; otherwise, the output is used as is. This also saves computation on answers that are already correct.

To quantify the verifier's impact, we analyze three distinct scenarios: *ProRefine (verifier)*, our standard approach which employs $LLM_{feedback}$ to guide refinement; *ProRefine (no verifier)*, wherein the refinement process operates without verifier input; and *ProRefine (optimal verifier)*, guided by a perfect verifier (simulated using ground-truth labels). This optimal condition reveals the upper bound of the refinement loop's potential. Consequently, the performance difference between *ProRefine (verifier)* and *ProRefine (optimal verifier)* underscores the significance of verifier accuracy. It is important to note that ProRefine's methodology does not inherently rely on labels or optimal verification, despite their use in this specific evaluation.

## 3.2 IMPLEMENTATION DETAILS

The granularity and duration of the optimization process are governed by two parameters: $k$, the number of tokens per step, and $n$, the maximum number of steps. These parameters can be adjusted based on the complexity of the task and the desired output quality. For example, rather than generating feedback every $k$ tokens, we might instead choose to provide feedback after each sentence or paragraph, particularly in tasks such as machine translation or text summarization, where larger semantic units may be more meaningful. We run all of our experiments on compute nodes with a single Nvidia A100 GPU (80GB), 24 core processor, and 220GB RAM. For model's `generate` function we set the following parameters: `num_return_sequences=1`, `do_sample=False`, `top_p=None`, `temperature=None`.

## 4 EXPERIMENTS AND EVALUATION

### 4.1 DATA

We evaluate ProRefine on five reasoning tasks, each of which involves multi-step reasoning, making them suitable for evaluating prompt optimization in agentic workflows. We include the original dataset split sizes in (train/validation/test) format: object counting and word sorting from the BIG-Bench Hard benchmark Srivastava et al. (2023) (50/100/100), grade-school math problem-solving from GSM8K Cobbe et al. (2021) (200/300/1319), math word problems from SVAMP Patel et al. (2021) (2516/622/1000), and algebraic word problems from AQUARAT Ling et al. (2017) (97467/254/254). We use the same splits and evaluation as Yuksekgonul et al. (2024) for object counting, word sorting, and GSM8K.

## 4.2 EXPERIMENTAL SETUP

| Dataset | Method | Llama-3.2 1B-it | Llama-3.2 3B-it | Llama-3.1 8B-it |
|---|---|---|---|---|
| Object Counting | CoT | 0.48 [0.382, 0.578] | 0.65 [0.556, 0.744] | 0.73 [0.643, 0.817] |
| | TextGrad | **0.62** [0.524, 0.716] | 0.73 [0.643, 0.817] | 0.86 [0.792, 0.928] |
| | ProRefine (no verifier) | 0.51 [0.412, 0.608] | **0.75** [0.665, 0.835] | 0.77 [0.687, 0.853] |
| | ProRefine (verifier) | 0.6 [0.503, 0.696] | 0.72 [0.632, 0.808] | **0.89**\* [0.839, 0.959] |
| | †ProRefine (optimal verifier) | 0.67 [0.577, 0.763] | 0.85\* [0.780, 0.920] | 0.94\* [0.893, 0.987] |
| Word Sorting | CoT | 0.11 [0.048, 0.172] | 0.10 [0.041, 0.159] | 0.50 [0.401, 0.598] |
| | TextGrad | **0.33**\* [0.237, 0.423] | **0.61**\* [0.514, 0.706] | 0.69\* [0.599, 0.781] |
| | ProRefine (no verifier) | 0.22 [0.138, 0.302] | 0.47 [0.372, 0.568] | 0.68 [0.595, 0.779] |
| | ProRefine (verifier) | 0.19 [0.113, 0.267] | 0.32\* [0.228, 0.412] | **0.71**\* [0.621, 0.799] |
| | †ProRefine (optimal verifier) | 0.29\* [0.192, 0.368] | 0.53\* [0.432, 0.628] | 0.86\*\* [0.792, 0.928] |
| GSM8K | CoT | 0.450 [0.423, 0.476] | 0.809 [0.787, 0.829] | 0.819 [0.797, 0.839] |
| | TextGrad | 0.463 [0.436, 0.489] | 0.801 [0.779, 0.822] | 0.864\* [0.845, 0.882] |
| | ProRefine (no verifier) | 0.636\*\* [0.610, 0.662] | 0.797 [0.774, 0.818] | 0.843 [0.823, 0.863] |
| | ProRefine (verifier) | **0.654**\*\* [0.627, 0.678] | **0.866**\*\* [0.847, 0.883] | **0.885**\* [0.868, 0.902] |
| | †ProRefine (optimal verifier) | 0.725\*\* [0.701, 0.749] | 0.904\*\* [0.888, 0.920] | 0.936\*\* [0.922, 0.949] |
| SVAMP | CoT | 0.689 [0.66, 0.718] | 0.869 [0.848, 0.890] | 0.854 [0.832 , 0.876] |
| | TextGrad | 0.684 [0.655, 0.713] | 0.861 [0.840, 0.882] | 0.84 [0.817, 0.863] |
| | ProRefine (no verifier) | 0.774\*\* [0.748, 0.800] | 0.878 [0.858, 0.898] | 0.877 [0.857, 0.897] |
| | ProRefine (verifier) | **0.808**\*\* [0.784, 0.832] | **0.896**\*\* [0.877, 0.915] | **0.893**\* [0.874, 0.912] |
| | †ProRefine (optimal verifier) | 0.861\*\* [0.840, 0.882] | 0.925\*\* [0.909, 0.941] | 0.938\*\* [0.923, 0.953] |
| AQUARAT | CoT | 0.259 [0.202, 0.31] | **0.563** [0.498, 0.620] | 0.586 [0.522, 0.643] |
| | TextGrad | **0.311** [0.250, 0.364] | 0.524 [0.462 , 0.585] | 0.559 [0.494, 0.616] |
| | ProRefine (no verifier) | 0.205 [0.151, 0.250] | 0.343 [0.284, 0.401] | 0.398 [0.337 , 0.458] |
| | ProRefine (verifier) | 0.268 [0.209, 0.318] | 0.551 [0.486 , 0.608] | **0.606** [0.542, 0.663] |
| | †ProRefine (optimal verifier) | 0.354 [0.292, 0.409] | 0.598 [0.538, 0.659] | 0.657 [0.595, 0.712 ] |

Table 1: Test Accuracy with 95% confidence intervals across five benchmark datasets and models. \* and \*\* denote statistically significant improvements over one or two baseline methods, respectively. Results in bold indicate the highest accuracy for a dataset-method combination. † demonstrates the upper bound potential of the optimization loop and the impact of verifier quality. *Llama3.1-70B-instruct* is employed for feedback generation, prompt optimization, and evaluation.

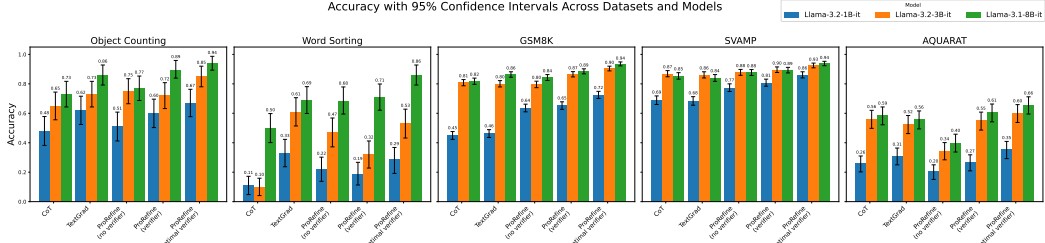

Figure 2: Test Accuracy [with 95% confidence interval] across different models and datasets. *Llama3.1-70B-instruct* is employed for feedback generation, prompt optimization, and evaluation.

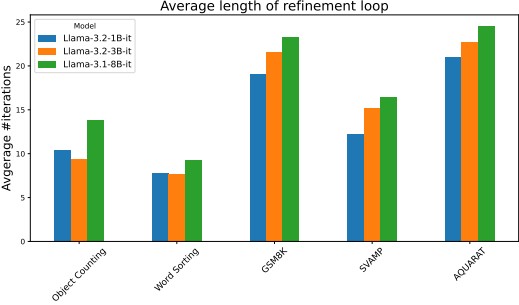

Figure 3: Average number of prompt refinement iterations.

We experiment with three models - *Llama3.2-1B-instruct*, *Llama3.2-3B-instruct*, and *Llama3.1-8B-instruct* Meta (2024) for $LLM_{task}$. The prompts are optimized using Algorithm 1, with *Llama3.1-70B-instruct* used for feedback generation, prompt optimization, and evaluation. We select the values

of hyperparameters $k = 10$ and $n = 25$ to control the granularity of feedback and duration of optimization. Hyperparameters $k$ and $n$ were fixed based on general preliminary exploration and not tuned per task using benchmark training/validation data.

We compare the performance of our method against the zero-shot Chain-of-Thought (CoT) baseline and TextGrad Yuksekgonul et al. (2024), and report test accuracy with 95% confidence interval. We choose TextGrad as a baseline because Yuksekgonul et al. (2024) reported performance at par or better than DSPy Khattab et al. (2024) for prompt optimization on object counting, word sorting, and GSM8k datasets. It is essential to remember that TextGrad is a supervised fine-tuning method that utilizes both the training and validation sets. For TextGrad, we use a comparative setup consisting of a task model to be fine-tuned and *Llama3.1-70B-instruct* model for feedback generation and backpropagation. The results are shown in Table 1 and Figure 2.

### 4.3 RESULTS

Our results (Table 1) demonstrate that ProRefine significantly improves $LLM_{task}$ performance over the zero-shot CoT baseline in all but one experiment, and it outperforms TextGrad in 11 out of 15 cases overall. For *Llama3.2-1B-instruct* model, ProRefine can significantly outperform CoT and TextGrad on 2 out of 5 datasets. For *Llama3.2-3B-instruct* model, ProRefine can outperform CoT and TextGrad on 3 out of 5 datasets with one significant result. For *Llama3.1-8B-instruct* model, ProRefine can outperform CoT and TextGrad on all 5 datasets with 4 significant results. TextGrad is a supervised baseline; ProRefine is training-free.

**Object Counting** ProRefine improves performance by $3 - 16$ percentage points over CoT, with significant gains observed for *Llama3.1-8B-instruct*. It outperforms TextGrad on 2 out of 3 models, yielding a $2 - 3$ percentage point advantage. However, a performance drop of 2 points is observed for *Llama3.2-1B-instruct*.

**Word Sorting** Performance gains over CoT range from $8 - 37$ percentage points, with significant improvements for *Llama3.2-3B-instruct* and *Llama3.1-8B-instruct*. ProRefine surpasses TextGrad on 1 of 3 models with a 2-point gain, but performance drops of $11 - 14$ points are observed for *Llama3.2-1B-instruct* and *Llama3.2-3B-instruct*.

**GSM8K** ProRefine achieves $2.4 - 20.4$ percentage points improvement over CoT, with significant improvement observed for all the models; however, a slight performance drop (1.2) is observed for *Llama3.2-3B-instruct*. It outperforms TextGrad on all models, achieving a $2.1 - 19.1$ percentage point gain with significant results observed for *Llama3.2-1B-instruct* and *Llama3.2-3B-instruct* models. Minor performance drop of $0.4 - 2.1$ is observed for *Llama3.2-3B-instruct* and *Llama3.1-8B-instruct*.

**SVAMP** Performance improves by $0.9 - 11.9$ percentage points over CoT, with significant gains for *Llama3.2-1B-instruct* and *Llama3.1-8B-instruct*. ProRefine outperforms TextGrad across all models, with $1.7 - 12.4$ percentage point gains and significant results for *Llama3.2-1B-instruct*.

**AQUARAT** Gains over CoT range from $0.9 - 2$ percentage points, but declines of $5.4 - 22$ points are also observed. ProRefine exceeds TextGrad on 2 of 3 models, with $2.7 - 4.7$ percentage point gains, though performance drops of $10.6 - 18.1$ points are also recorded.

Our results demonstrate that using ProRefine with an optimal verifier significantly improves performance for all tasks, achieving the best results in 13 out of 15 cases, highlighting the critical role of verifier quality. Notably, number of significant improvements increases with larger model sizes.

We also observe that ProRefine enables smaller models, such as *Llama3.2-3B-instruct* and *Llama3.1-8B-instruct*, to approach the zero-shot performance of larger models like *Llama3.1-8B-instruct* and *Llama3.1-70B-instruct*, respectively. In our experiments using *Llama3.1-70B-instruct* for $LLM_{task}$, some experiments yielded an accuracy of 1, suggesting potential data contamination. We also experimented using the full chat history at each step, but it failed to produce consistent improvements over our stateless approach. Additionally, we experimented using ProRefine with a comparatively smaller model (*Llama3.1-8B-instruct*) for $LLM_{feedback}$ and $LLM_{optimizer}$. However, this configuration also failed to consistently yield substantial improvements and, in some cases, even degraded perfor-

mance. This aligns with prior findings indicating that smaller models are less capable of providing high-quality feedback Saunders et al. (2022); Bai et al. (2022).

## 5 DISCUSSION

Regarding RQ1, the results demonstrate that ProRefine is a broadly applicable method that utilizes textual feedback to improve LLM performance at inference time. The "performance gap bridging" effect is particularly noteworthy, suggesting that ProRefine may serve as an effective alternative to simply scaling up model size, potentially avoiding costly fine-tuning an advantage in resource-constrained settings.

The largest performance gains are observed on the word sorting task, indicating that tasks requiring more complex reasoning or manipulation of intermediate outputs benefit the most from ProRefine's iterative refinement. The mixed results when using a smaller model for $LLM_{feedback}$ illustrate the importance of "knowledge asymmetry," i.e., that the feedback model should be "sufficiently capable" of providing useful critiques.

Regarding RQ2, the results indicate that ProRefine outperforms the baselines on 2 and 3 datasets when using the *Llama3.2-1B-instruct* and *Llama3.2-3B-instruct* models, respectively, and on all 5 datasets when using the *Llama3.1-8B-instruct* model. This suggests that performance improvements scale with model size. These findings imply that larger models are preferable to smaller ones, particularly in agentic workflows that may require test-time scaling and the effective use of textual feedback to solve complex tasks.

Regarding RQ3, the results highlight that employing a high-quality verifier is crucial for significantly improving task performance at inference time. We observe some cases where "no verifier" outperforms the "verifier" setting, which indicates the verifier incorrectly accepted a flawed initial answer, thereby preventing the refinement process from correcting the error. This reveals a trade-off: the verifier reduces computational cost on correct answers but risks prematurely halting on incorrect ones. The superior results of the "optimal verifier" highlight the critical role of verifier accuracy. Beyond enhancing performance, the verifier also reduces computational cost during inference by guiding the refinement process. Moreover, it opens up promising avenues for future work, where an optimizer could be designed to maximize rewards guided by the verifier's assessments. ProRefine can offer a degree of interpretability by exposing the outputs from $LLM_{feedback}$, allowing insights into the model's reasoning process. Figures 4 and 5 demonstrate cases where model feedback successfully improves the output and where it fails, respectively. Although evaluated on reasoning and math tasks, ProRefine is general and applicable to other tasks.

A crucial consideration for ProRefine is the trade-off between its accuracy gains and the increased computational cost at inference time. Each refinement step requires additional calls to the $LLM_{feedback}$ and $LLM_{optimizer}$, making any single query more expensive to process than a standard single-pass generation. However, this per-query cost should be evaluated within ProRefine's intended hybrid-model deployment. The strategy is not to run refinement on every query, but to use it as an on-demand intervention precisely when a more efficient base model fails. Therefore, the overall system cost is not a simple sum of expensive refinement processes. Instead, it is a blend of low-cost successes from the base model and high-value, targeted corrections. Moreover, the cost is still considerably lower than full model retraining or fine-tuning. Our results support this approach's practicality: Figure 3 shows that the average number of refinement iterations is typically low, ensuring the per-incident cost of intervention is contained. This cost-accuracy balance can be further optimized by tuning hyperparameters like feedback granularity ($k$) and maximum iterations ($n$).

## 6 LIMITATIONS AND FUTURE WORK

This work has the following limitations that we acknowledge have potential for future explorations:

- **Computational Cost and Practicality:** While ProRefine is designed for cost-effective hybrid deployments, its iterative process inherently increases inference-time latency and computational cost compared to a single-pass query. The cost-benefit of this trade-off must be carefully evaluated for each specific application, as its viability depends on the base model's failure rate and the relative costs of the LLMs involved.

- **Generalizability:** Our evaluation is currently focused on mathematical and multi-step reasoning tasks. Further research is needed to assess performance across a broader range of reasoning tasks and domains. Our method is also sensitive to hyperparameters and requires manual tuning. Developing more robust, automated, or adaptive methods for setting parameters would enhance the method's usability.

- **Dependence on High-Quality Feedback:** The system's performance is dependent on the quality of the $LLM_{feedback}$. Future work could explore using a specialized "critic" model or fine-tuning feedback models to improve diagnostic accuracy. Furthermore, using LLMs for evaluation introduces potential biases and more comprehensive human evaluations and robust methods are need for mitigating evaluator bias.

- **Stability of the Refinement Loop:** The iterative nature of ProRefine lacks a formal convergence guarantee. In some cases, the refinement process can suffer from prompt degradation after many iterations or plateau before reaching an optimal solution. Investigating methods to ensure stable and monotonic improvement is a key area for future research.

## 7 CONCLUSION

We introduced ProRefine, a novel, practical, and *inference-time* prompt optimization method for agentic workflows. ProRefine leverages LLM-generated textual feedback to dynamically refine prompts, leading to significant performance improvements on multi-step reasoning tasks without requiring additional training or ground-truth labels. Our results demonstrate its ability to bridge the performance gap between smaller and larger LLMs, making it a key enabler for more efficient and cost-effective hybrid-model deployments. The *inference-time* nature of ProRefine makes it readily deployable for on-demand reasoning correction, contributing to more adaptable and accessible AI systems. Future work will explore applying this framework to new domains, developing more sophisticated feedback and optimizer agents, and exploring adaptive policies for hyperparameter tuning to further optimize the cost-performance trade-off.

## REPRODUCIBILITY STATEMENT

We have taken several steps to ensure the reproducibility of our results. The main paper provides a detailed description of the ProRefine framework, including algorithmic steps (see Section 3), experimental setup (Section 3.2), and evaluation protocols. Hyperparameter choices and implementation details are further elaborated in Section 3. All datasets used in our experiments are publicly available, and we include a comprehensive summary of data processing steps in the supplementary materials. To facilitate replication, we have included anonymized source code and evaluation scripts as part of the supplementary submission. We will release the source code along with the final submission once accepted for publication.

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

## A SUPPLEMENTARY MATERIAL

### A.0.1 LLM USAGE

LLMs (ChatGPT 4 and Gemini 2.5) were used to aid in grammar and writing flow.

### A.1 EXPERIMENTAL CODE

Experimental code available via `https://anonymous.4open.science/r/prorefine-5525/README.md`

## A.2 ADDITIONAL EXAMPLE #1

**Input (except for Answer)**

**Initial Prompt:** You will answer a reasoning question. Think step by step. The last line of your response should be of the following format: 'Answer: $VALUE' where VALUE is a numerical value.

**Query:** I have a lettuce head, a stalk of celery, two yams, a cauliflower, a carrot, two onions, two potatoes, and three cabbages. How many vegetables do I have?
**(Answer:** 13)

**Baseline response (no ProRefine)**

**LLM$_{task}$:** To determine the number of vegetables, we need to identify the vegetables in the given list. \n\n- Lettuce is a vegetable.\n- Celery is a vegetable.\n- Cauliflower is a vegetable.\n- Carrot is a vegetable.\n- Onion is a vegetable.\n- Potato is a vegetable.\n- Cabbage is a vegetable.\n\nNow, let's count the number of vegetables: \n1. Lettuce\n2. Celery\n3. Cauliflower\n4. Carrot\n5. Onion\n6. Potato\n7. Cabbage\n\nThere are 7 vegetables in total.\n\nAnswer: 7

**ProRefine**

**LLM$_{task}$:** *To determine the number of vegetables*

**LLM$_{feedback}$:** *The output is incomplete, it should directly answer the question. Add the total count of the vegetables: "There are 12 vegetables."*

**LLM$_{Optimizer}$:** *You will answer a reasoning question. Think step by step. Ensure the answer directly answers the question and provides a final numerical value. If necessary, include the total count of the vegetables in the answer. The last line of your response should be of the following format: \'Answer: $VALUE\' where VALUE is a numerical value. Include the total count of the vegetables in the format "There are X vegetables."*

**LLM$_{task}$:** *To find the total number of vegetables, we need to count each type of vegetable*

**LLM$_{feedback}$:** *Incorrect assumption. The task is to find the total number of individual vegetables, not just count the types. Instead, focus on counting the quantities of each vegetable and summing them up.*

**LLM$_{Optimizer}$:** *You will answer a reasoning question. Think step by step. To find the total count of individual vegetables, count the quantities of each vegetable separately and sum them up. Then, provide the final numerical value. The answer should include the total count of individual vegetables, not just the types. Include the total count in the format "There are X individual vegetables." The last line of your response should be of the following format: \'Answer: $VALUE\' where VALUE is a numerical value.*

**LLM$_{task}$:** *To find the total count of individual vegetables, we need to count the quantities of each vegetable separately and sum them up.\n\n- Let*

**LLM$_{feedback}$:** *Incorrect calculation, the output should directly provide the total count of vegetables instead of explaining the process. It should correctly sum up the quantities of each vegetable, which is 1+1+2+1+1+2+2+3=13.*

**LLM$_{Optimizer}$:** *You will answer a reasoning question. To find the total count of individual vegetables, you are given the quantities of each vegetable. Directly provide the total count of individual vegetables by summing up the quantities. Do not explain the process. Ensure the answer includes the total count in the format "There are X individual vegetables." The last line of your response should be of the following format: \'Answer: $VALUE\' where VALUE is a numerical value representing the total count of individual vegetables.*

**LLM$_{task}$:** *There are 1 + 1 + 2 + 1 + 1 + 2 + 2 + 3 = 13 individual vegetables.\n\nAnswer: 13*

**LLM$_{feedback}$:** the output is correct ✅

Figure 4: ProRefine example. Given an input query (which in this case has a correct answer: 13) and an initial prompt, the task model ($LLM_{task}$) gives an incorrect answer. ProRefine uses two additional models, $LLM_{feedback}$ and $LLM_{optimizer}$, to iteratively improve the prompt *as the $LLM_{task}$ generates its response*. Refining the prompt during generation allows the feedback model to target local regions of the response, providing finer-grained feedback than waiting for the response to complete. We also provide an additional example illustrating our approach in Figure 5.

## A.3 Additional Example #2

**Input (except for Answer)**

**Initial Prompt:** You will answer a reasoning question. Think step by step. The last line of your response should be of the following format: 'Answer: $VALUE' where VALUE is a numerical value.

**Query:** Sort the following words alphabetically: List: oakland seaborg jacobi membrane trapezoidal allis marmot toggle anthology
**(Answer:** allis anthology jacobi marmot membrane oakland seaborg toggle trapezoidal)

**Baseline response (no ProRefine)**

**LLM$_{task}$:** To sort the given words alphabetically, we need to compare each word and arrange them in order from A to Z.\n\nHere's the step-by-step process:\n\n1. Compare 'allis' and 'anthology'. 'allis' comes before 'anthology'.\n2. Compare 'anthology' and 'jacobi'. 'anthology' comes before 'jacobi'.\n3. Compare 'jacobi' and 'list'. 'jacobi' comes before 'list'.\n4. Compare 'list' and 'marmot'. 'list' comes before 'marmot'.\n5. Compare 'marmot' and 'membrane'.'marmot' comes before 'membrane'.\n6. Compare 'membrane' and 'oakland'.'membrane' comes before 'oakland'.\n7. Compare 'oakland' and 'seaborg'. 'oakland' comes before 'seaborg'.\n8. Compare 'seaborg' and 'trapezoidal'.'seaborg' comes before 'trapezoidal'.\n\nNow, let's arrange the words in alphabetical order:\n\n1. allis\n2. anthology\n3. jacobi\n4. list\n5. marmot\n6. membrane\n7. oakland\n8. seaborg\n9. trapezoidal\n\nAnswer: 9

**ProRefine**

**LLM$_{task}$:** To sort the given words alphabet

**LLM$_{feedback}$:** Incomplete output. The output should provide the sorted list of words instead of just stating the instruction. \n\nCorrect output should list the words in alphabetical order, such as: allis, anthology, jacobi, marmot, membrane, oakland, seaborg, trapezoidal, toggle

**LLM$_{Optimizer}$:** You will answer a reasoning question that requires sorting a list of words. The input will be a list of words, and you need to think step by step to provide the correct output. Your response should include the sorted list of words in alphabetical order. Do not forget to list all the words in the correct order. Think step-by-step and provide the correct output.\n\nThe last line of your response should be of the following format: 'Answer: $VALUE' where VALUE is the sorted list of words, with each word separated by a comma. For example: 'Answer: allis, anthology, jacobi, marmot, membrane, oakland, seaborg, trapezoidal, toggle'

**LLM$_{task}$:** To sort the list of words alphabetically, I will go through each word and

**LLM$_{feedback}$:** This is not the correct output. The task was to sort the list of words alphabetically, but the given output starts with an explanation. A correct output should be a direct response with the sorted list of words. \n\nHowever, the output did not provide the sorted list. The correct sorted list should be: allis, anthology, jacobi, marmot, membrane, oakland, seaborg, toggle, trapezoidal.

**LLM$_{Optimizer}$:** You will answer a reasoning question that requires sorting a list of words. The input will be a list of words, and you need to sort the list in alphabetical order and provide the correct output directly. Do not include any explanations. \n\nYour response should be a direct list of the sorted words in alphabetical order, separated by commas. Ensure that all words are included in the correct order. \n\nThe last line of your response should be of the following format: 'Answer: $VALUE' where VALUE is the sorted list of words, with each word separated by a comma.

**LLM$_{task}$:** Answer: allis, anthology, jacobi, list, marmot, membrane, oakland, seaborg, trape

**LLM$_{feedback}$:** Missing word "toggle" in the sorted list. Also, the word "list" was not supposed to be included as it was part of the original instruction, not the actual list to be sorted. \n\nCorrected sorted list should be: allis, anthology, jacobi, marmot, membrane, oakland, seaborg, toggle, trapezoidal.

**LLM$_{Optimizer}$:** You will answer a reasoning question that requires sorting a list of words. The input will be a list of words, and you need to sort the list in alphabetical order and provide the correct output directly. Do not include any explanations. Ensure all words from the input list are included in the sorted output.\n\nYour response should be a direct list of the sorted words in alphabetical order, separated by commas. The last line of your response should be of the following format: 'Answer: $VALUE' where VALUE is the sorted list of words, with each word separated by a comma.

**LLM$_{task}$:** Answer: allis, anthology, jacobi, list, marmot, membrane, oakland, seaborg, trapezoidal, toggle

**LLM$_{feedback}$:** "Correct alphabetization, but \'list\' is not part of the original list, remove it to maintain consistency." ✗

**LLM$_{Optimizer}$:** You will answer a reasoning question that requires sorting a list of words. The input will be a list of words, and you need to sort the list in alphabetical order and provide the correct output directly. Do not include any explanations. Ensure all words from the input list are included in the sorted output, and do not add any extra words. \n\nYour response should be a direct list of the sorted words in alphabetical order, separated by commas. The last line of your response should be of the following format: 'Answer: $VALUE' where VALUE is the sorted list of words, with each word separated by a comma.

Figure 5: This figure is an instance when $LLM_{optimizer}$ is not aligned with the feedback from $LLM_{feedback}$ and misses important guiding steps. The framework is similar to Figure 4 when $LLM_{optimizer}$ conveys feedback effectively. We've observed a few failed instances following this pattern.

