# OpenReview forum: "ProRefine: Inference-time Prompt Refinement with Textual Feedback"
_ICLR.cc/2026/Conference — ICLR 2026 Conference Withdrawn Submission_

### Official Review · Reviewer_HKpp · 2025-10-17

**Soundness:** 2
**Presentation:** 2
**Contribution:** 2
**Rating:** 2
**Confidence:** 3

**Summary:**

The paper proposes ProRefine, a per-instance test-time method to perform prompt optimization. The method iteratively enhances a "task" LLM with a "feedback" LLM, which critiques a partial model output, and an "optimizer" LLM, which refines the input prompt for the next iteration. An optional verifier is also employed for early termination. The authors experiment with 3 different Llama-3 variants of different sizes under a feedback and optimizer (and verifier) models on 5 benchmarks.

**Strengths:**

- The progressive continuation trick (i.e., using $i*k$ tokens per round $i$) seems like an interesting to catch and correct mistakes early in generation; However, the effects of this are not ablated (see weaknesses)
- ProRefine appears to work decently in a few settings, e.g., +21% on Word Sorting for Llama-3.1-8B.

**Weaknesses:**

- ProRefine relies on larger models to provide feedback and optimize the prompt.
  - The authors bring up that they are concerned with resource-constrained environments where querying a capable feedback model is feasible. But is this realistic? Under what scenario would a practitioner be able to call a larger more capable LLM on-demand, but *only for feedback*?
>It is designed for resource-constrained environments where deploying the largest models for every query isn’t feasible, but temporary access to a capable feedback LLM (perhaps via a separate API call) is possible for critical tasks. In such cases, the refinement process is triggered as an on-demand “expert intervention."
  - Authors contend that external costs to optimizers/feedback models should not be considered. However, I do not see any proof of the existence of instances where this feedback/optimizer loop is **not** triggered. In fact, many benchmarks seem to trigger the maximum number of refinement rounds **on average** (e.g., AQUARAT for Llama-3.1-8B). As such, there's no demonstration that such a hybrid model strategy (1) is realistic and (2) actually operates as desired, e.g., only appeals to larger LLMs a small enough portion of the time such that costs can be ignored
>Each refinement step requires additional calls to the LLM feedback and LLM optimizer , making any single query more expensive to process than a standard single-pass generation. However, this per-query cost should be evaluated within ProRefine’s intended hybrid model deployment. The strategy is not to run refinement on every query, but to use it as an on-demand intervention precisely when a more efficient base model fails.
  - The authors note that experiments with smaller models (Llama-3.1-8B-Instruct) failed to substantially improve performance, with degradations in some instances. This raises a key question: If ProRefine operates on a per-instance level, and requires multiple model calls to a larger model, why not simply use the larger model for inference? For example, ProRefine+Llama-3.1-70B-as-verifier with Llama-3.1-8B as the task model on GSM8K achieves 88.5%. Llama-3.1-70B self-reports 95.1% (8-shot) performance with CoT. Other papers have reported similar performance with 0-shot prompting (e.g., https://arxiv.org/abs/2504.15253, Table 3).
- Insufficient baselines: A purported benefit of ProRefine is that it is a training-free, inference-time solution. As such, I believe that it should be compared to other training-free, inference-time methods, such as debate, self-consistency, best-of-N with external verifier. It is important to demonstrate that ProRefine outperforms other methods in this category.
- Performance improvement given extra compute is not always convincingly better: While there are some instances where ProRefine beats CoT and TextGrad by significant margins, there are others where a lot of inference-time compute is burned without much to show for it. Per Fig 3, ProRefine can make up to 25 rounds of prompt refinement (The max number of rounds set by authors; this indicates that more rounds could be possible if left uncapped). On AQUARAT, this 25 rounds of refinement yields ~2 absolute percentage increase over a vanilla CoT baseline, which requires a single model inference call. This seems like a lot of compute to spend per instance. I am not convinced that the trade-off is worth it
- Insufficient ablations: Authors do not probe the performance as a function of token chunks $k$ or number of refinement rounds $n$.
- Questions about potential contamination: The authors report that Llama-3.1-70B achieves 100% accuracy on some tasks, and attribute this to potential data contamination. If authors believe this to be true, why are they comfortable using Llama-3.1-70B as a verifier? If contamination is not the cause, this reinforces my first weakness.
>In our experiments using Llama3.1-70B-instruct for LLMtask, some experiments yielded an accuracy of 1, suggesting potential data contamination.

**Questions:**

- Do you have a per compute (e.g., FLOPS) comparison between ProRefine and TextGrad (and other inference-time methods)? One can approximate inference time FLOPS using well-known formulas, e.g., like that in https://arxiv.org/abs/2408.03314. As TextGrad "training" is purely inference driven, one can quantify total number of FLOPS required to optimize the prompt. It would be interesting to see which methods drive the most improvement per compute spent.
- Some suggestions:
  - It appears that Fig 2 repeats information already presented in Table 1. I would replace it with additional experiments, e.g., those suggested above or in weaknesses.
  - There's some ambiguity in the $k$ token mechanism. My first impression from Fig 1 caption was that all rounds share a continued generation prefix of $i*k$ tokens ("Each refinement iteration updates the prompt for future tokens; previous tokens remain unchanged."). However, the algorithm and examples in the appendix seem to clarify that the entire generation changes iteration by iteration, only the generation is allowed more inference-time tokens.

---

### Official Review · Reviewer_LcqN · 2025-10-31

**Soundness:** 2
**Presentation:** 2
**Contribution:** 2
**Rating:** 2
**Confidence:** 4

**Summary:**

The paper introduces ProRefine, an innovative method for optimizing prompts during inference in LLMs. It leverages textual feedback generated by LLMs to dynamically refine prompts, enhancing the models' performance on multi-step reasoning tasks without the need for additional training or ground-truth labels.

**Strengths:**

1. ProRefine presents a novel approach to prompt optimization during inference, distinguishing itself from existing methods by utilizing LLM-generated textual feedback for dynamic refinement. This innovative use of feedback not only enhances the reasoning capabilities of LLMs but also addresses the limitations of prior techniques that often rely on extensive training data or fixed prompts.

2. The authors provide a comprehensive evaluation of ProRefine across multiple reasoning tasks, demonstrating its effectiveness compared to established baselines like Chain-of-Thought and TextGrad. The results are presented with clarity, supported by empirical evidence that highlights the method's advantages in various scenarios.

**Weaknesses:**

1. The paper does not clearly state the additional cost and latency introduced by the proposed method. Please provide quantitative results or analysis to clarify this aspect.

3. The experiments primarily evaluate relatively small and weak open-source models (mainly LLaMA). It remains unclear whether the conclusions generalize to larger models (e.g., those exceeding 30B parameters) or to more capable closed-source models.

3. I also wonder whether these tasks might already be relatively easy for advanced models. If state-of-the-art models can already perform well on them, how much additional value does the proposed method provide?

**Questions:**

See above

---

### Official Review · Reviewer_GGQH · 2025-11-01

**Soundness:** 3
**Presentation:** 3
**Contribution:** 2
**Rating:** 4
**Confidence:** 4

**Summary:**

ProRefine is an inference-time prompt-refinement method that improves LLM reasoning without any model training or ground-truth supervision. The approach forms a closed loop between three roles: LLMtask, which generates partial outputs; LLMfeedback, which critiques the intermediate reasoning; and LLMoptimizer, which uses this feedback to refine the prompt for subsequent decoding steps. The model generates only a few tokens per step (k=10) and iteratively updates the prompt for up to 25 iterations, leaving previously generated tokens unchanged.

The authors evaluate ProRefine across five reasoning benchmarks, object counting, word sorting, GSM8K, SVAMP, and AQUARAT. They compare it to zero-shot CoT	prompting and the supervised TextGrad baseline. Across all tasks, ProRefine yields consistent performance gains, improving accuracy over zero-shot CoT by 3 to 37 percentage points, and outperforming TextGrad in 11 out of 15 model-dataset combinations. The technique shows particularly large improvements on tasks requiring complex manipulation of intermediate outputs, such as sorting, and benefits more as the base model size increases.

**Strengths:**

1.	A training-free, label-free method, ProRefine improves reasoning at inference time with textual feedback, no requirement for fine-tuning; suitable for black-box LLMs.

2.	Analysis showing how feedback quality affects performance: Comparing no verifier, verifier, and an optimal verifier shows the method’s upper bound and the centrality of verifier quality; the optimal verifier yields best results most of the times.

3.	Paper positions ProRefine for on-demand use in hybrid systems and reports that the average number of refinement iterations is low (Fig. 3), supporting practical latency/cost trade-offs.

**Weaknesses:**

The key challenge I have with the paper is that it's not well positioned to the current literature. This makes the novelty of the paper unclear. I would urge the authors to compare related SOTA and even evaluate them against ProRefine. The proposed approach has been applied by previous works and what is unique contribution in this paper is not clear.

1.  The evaluations and comparison to SOTA is very weak. The authors compare to just Textgrad, there are several other prompt optimization techniques such as DsPy, PromptWizard, PromptBreeder, etc. comparison to SOTA and positing the work is missing.

2. Strong dependence on verifier quality: The paper shows the optimal verifier produces the best result in several model×dataset cases, and explicitly notes cases where a real verifier can incorrectly accept a flawed answer and block refinement.

3. Given the reliance of LLM at every stage, t is very critical to compare the cost, latency and tokens of the proposed approach to SOTA.

4. Sensitivity to hyperparameters (k, n) and fixed choices: Method granularity/duration is controlled by tokens-per-step k and max-steps n; the authors fixed k=10, n=25 based on preliminary exploration. Authors fail to discuss variations of these and how these were selected.

**Questions:**

1. Describe and evaluate Prorefine with other SOTA approaches
2. Clearly provide results on latency, tokens used, API calls of prorefine.
3. Discuss hyper parameters used and show ablations.

---

### Note · Authors · 2025-11-18

**Comment:**

We would like to sincerely thank the reviewers for the time and effort invested in evaluating our submission.

**Withdrawal Confirmation:**

I have read and agree with the venue's withdrawal policy on behalf of myself and my co-authors.